# PolyMeme: Fine-Grained Internet Meme Sensing

**DOI:** 10.3390/s24175456

**Published:** 2024-08-23

**Authors:** Vasileios Arailopoulos, Christos Koutlis, Symeon Papadopoulos, Panagiotis C. Petrantonakis

**Affiliations:** 1School of Electrical & Computer Engineering, Aristotle University of Thessaloniki, 54124 Thessaloniki, Greece; arailosb@gmail.com (V.A.); ppetrant@ece.auth.gr (P.C.P.); 2Information Technologies Institute @ Centre for Research and Technology Hellas, 57001 Thessaloniki, Greece; papadop@iti.gr

**Keywords:** meme taxonomy, meme classification, meme detection

## Abstract

Internet memes are a special type of digital content that is shared through social media. They have recently emerged as a popular new format of media communication. They are often multimodal, combining text with images and aim to express humor, irony, sarcasm, or sometimes convey hatred and misinformation. Automatically detecting memes is important since it enables tracking of social and cultural trends and issues related to the spread of harmful content. While memes can take various forms and belong to different categories, such as image macros, memes with labeled objects, screenshots, memes with text out of the image, and funny images, existing datasets do not account for the diversity of meme formats, styles and content. To bridge this gap, we present the PolyMeme dataset, which comprises approximately 27 K memes from four categories. This was collected from Reddit and a part of it was manually labelled into these categories. Using the manual labels, deep learning networks were trained to classify the unlabelled images with an estimated error rate of 7.35%. The introduced meme dataset in combination with existing datasets of regular images were used to train deep learning networks (ResNet, ViT) on meme detection, exhibiting very high accuracy levels (98% on the test set). In addition, no significant gains were identified from the use of regular images containing text.

## 1. Introduction

The evolution of social media has given rise to new forms of communication, with that of Internet Memes being one of the most popular, capturing the attention of countless users worldwide [1]. Memes can take various forms such as images, GIFs, videos or sometimes plain text [2]. Most frequently they are multimodal, combining text with images, and convey humorous or relatable content. They often contain references to current events and viral phenomena [3]. Memes can often be harmful and contribute to the spread of hate speech and misinformation. This has created the need for (semi-) automatically moderating this type of content [4,5]. To this end, a first necessary step involves robust (to all meme types) methods for “sensing” meme content [6,7] from large streams of social media posts.

An image meme most often comprises an image superimposed with text. There is diversity in the forms that image memes can take, but most of them seem to adhere to specific patterns, based on the presence, position and style of the superimposed text, as we can see in Figure 1. Depending on the pattern they follow, they can be considered to belong to a particular category of meme images, such as image macro, object labeling, screenshot, text out of image, and funny image [8]. Currently available image meme datasets are limited. To our knowledge, none differentiate between various meme types and formats, as detailed in Section 3.

Additionally, memes are special image cases involving subtle differences in terms of the background image semantics as well as the overlay text font appearance and position. These peculiar cues, which differentiate them from the images of standard image recognition tasks, render simple transfer learning less powerful and backbone fine-tuning is deemed beneficial for the task of meme detection and classification. Hence, the datasets used for such tasks should be of large size in order to enable the backbones to learn the corresponding patterns and adequately generalize. However, current available datasets are of limited size, as shown in Table 1.

In this work, we first provide a taxonomy for the main categories of image memes based on their format and use it to curate an extensive collection of images that adequately covers the taxonomy. The introduced PolyMeme dataset comprises approximately 27 k image memes, collected from the social media platform Reddit https://www.reddit.com/ (accessed on 4 June 2024) and annotated with one of the proposed categories per image. The collection process included the identification of popular Reddit groups (subreddits) pertinent to the meme types of interest, as well as content crawling, filtering and deduplication. The annotation process was semi-automatically performed by a Deep Learning (DL) model trained on a manually annotated subset (4 k images) of PolyMeme. A comparison of PolyMeme with existing meme datasets can be found in Table 1.

The main contributions of this work are the following:We provide a taxonomy of different image meme types.We introduce a large diverse dataset in terms of meme type.We establish a new benchmark for the meme detection task.We train state-of-the-art meme detection models exhibiting high accuracy levels.

## 2. Related Work

The main problems encountered by the research community concerning image memes address three main tasks: (a) distinguishing memes from regular images, also known as meme detection [7,11,12,13,14], (b) identifying hate speech in memes [5,15,16,17,18] and (c) clustering or categorizing memes in various ways [19,20]. These studies mainly aim to train deep learning models to solve the aforementioned tasks. Most of them consider deep Convolutional Neural Networks (CNNs), such as AlexNet, VGG, ResNet and EfficientNet or Transformer architectures such as ViT, for the processing of the meme’s visual part, and deep text embedding models, such as BERT, GloVe and FastText, for the processing of the meme’s textual part.

Existing meme datasets fail to provide a clear distinction of the different formats that memes follow. Previous studies on meme detection either overlook using a dataset that utilizes a wide range of meme formats or they often fall short in terms of dataset size. For instance, the study in [7] performs the task on a dataset, which includes images belonging solely to one format, while the study in [6] uses a dataset with more than one format but of relatively small size.

The Facebook Hateful Memes [4] and Memotion 7 k [9] datasets each contain 10 k text annotated images, with the former including exclusively memes of the format known as Image Macros https://en.wikipedia.org/wiki/Image_macro, accessed on 4 June 2024. The purpose of these datasets is to address the task of detecting hate speech or sentiment in memes. The MultiOFF [5], DankMemes [6] and SemanticMemes [13] are some of the datasets that cover more formats, but their sizes are significantly smaller compared to that of Facebook Hateful Memes and Memotion. MemeCap [10] is a dataset oriented to meme captioning and has a size of 6.3 k images with multiple formats. Other studies [14,21] rely on web-scraped images from media platforms, such as Twitter, Reddit, Tumblr, Google Images, Facebook and Instagram. Table 1 provides a summary of existing meme datasets.

## 3. Polymeme Dataset

### 3.1. Meme Categories

Based on previous studies, we identified five meme categories with respect to differences in terms of morphology. This refers to the placement and structure of the annotated text on the image or even its absence. We present pertinent examples in Figure 1.

**Image Macros.** This is a category of images with well-defined characteristics. It has been the subject of analysis in several studies [2,4,22,23,24]. Its primary features are the text font and placement. The text predominantly occupies the upper and/or lower part of the image, and the font used is the Impact Font with black border and white capital letters. Their categorization is based more on the text’s positioning within the image, regardless of whether the font used is Impact or not.

**Object Labeling.** This comprises images where the text does not have a default position. Instead, it is placed upon objects or individuals and is used as a means of labeling them. As suggested in [1], a format of memes appeared that seem to ‘flood the image with text’ and is a common technique recognized and used by the online community to create new memes https://knowyourmeme.com/memes/object-labeling, accessed on 4 June 2024. This category can also be characterized by adding text where there is a natural blank space in the image or by altering existing text with another word or phrase.

**Screenshots.** A relatively distinct category of images that has gained significant popularity recently, consists of images originating from a screenshot of a tweet [3,25]. Characteristic of these images is the inclusion of the profile picture, the username, and the name of the user preceded by the “@” symbol (known as the tweet handle) before the main body of the post. However, we can extend this category to include screenshots from any source, whether they are messages, posts, or comments on other websites.

**Text out of Image.** In [25], attention was drawn to a category of memes where textual content is located outside of the image. By examining Internet meme images, it became apparent that this morphology is quite widespread. In general, this category includes images with text positioned outside the image area, over a white or dark background.

**Funny Images.** In [8], two categories of memes are mentioned: “reaction photoshops” and “photo fads”. These contain minimal to no text and, in contrast to the other categories, do not depend on textual content to convey their meaning. They are based on photographs that were edited in such a way as to recreate a humorous situation or depict viral Internet phenomena. Other instances belonging to this category are everyday images, captured with a camera, that showcase funny, interesting, or peculiar situations meant to capture the viewer’s attention.

### 3.2. Data Collection

The dataset was collected from Reddit that comprises groups where meme content is widely shared. A thorough investigation was made for subreddits that are widely popular and contain memes mainly in the form of images. The data were collected using the RipMe app https://github.com/RipMeApp/ripme, accessed on 4 June 2024, through which we were able to identify the top upvoted posts per subreddit, filtered based on different time criteria: top submissions of all time, year and month. In total, 27,881 URLs were gathered, from the total 38,863 links, of which 4402 were files other than images, and the remaining 6580 were duplicates. In the process of image downloading, there were several failure cases, where photos could not be downloaded because the links were broken or the images were no longer available. Hence, a total of 27,675 images were finally downloaded. To empirically assess the composition of the collected dataset, we manually annotated a random sample of 500 images to determine their category. During this check, we found memes of the defined categories (91.6%), memes that belong to multiple categories (5%), memes that did not definitively fall into any of the defined categories (1.4%), and even non-meme images (2%). The results of this annotation are shown in Table 2.

### 3.3. Data Pre-Processing

We used the fdupes program https://github.com/adrianlopezroche/fdupes, accessed on 4 June 2024, which operates by comparing the sizes of the files and then making a byte-to-byte comparison, to remove 189 exact duplicates. Additionally, we used the difference hashing method (dhash https://github.com/benhoyt/dhash, accessed on 4 June 2024) to identify and remove another 223 images that were either cropped or their resolution was changed. We discarded the funny images class, that contained 349 samples, from the dataset and the consequent analysis due to the following reasons:Only 0.4% of the collected data fall into this category (cf. Table 2).There is no way to search for this type of meme without starting from a list of specific examples in other sources (e.g., Twitter/X, Pinterest, Google Images).It is continuously updated by current events, and there is no specific pattern that it follows such as other categories where the textual content plays a significant role.

We also discarded images that do not contain any text since these images are most probably regular images and certainly they do not belong to any of the defined meme categories. To this end, we made use of the state-of-the-art text detector TextFuseNet [26] to identify and discard 249 such images.

### 3.4. Data Annotation

We annotated each image of the collected dataset with one of the four categories, namely Image Macros, Object labeling, Screenshots, and Text out of Image. The total number of images is approximately 27 K. Thus, manually annotating the whole dataset, a demanding and time-consuming task, was deemed infeasible. To address this issue, we considered machine learning techniques, which can be employed to semi-automatically categorize the images.

First, for each of the four categories, 1000 random samples were manually annotated. Then, following [27], we labelled the unlabeled part of the dataset using a classifier trained on the manually labeled part of it. To obtain the most accurate data annotation, we considered several architectures and two classification approaches. Specifically, the tested architectures include: (i) a simple CNN built from scratch (Toy CNN), three different state-of-the-art CNN models pretrained on ImageNet-1k, namely (ii) VGG16, (iii) ResNet152 and (iv) EfficientNetB4, and (v) the base version of the Visual Transformer (ViT) model pretrained on ImageNet-21k. We have made these architecture choices based on the fact that they have demonstrated state-of-the art performance on numerous computer vision tasks and have proven to be easily adaptable through fine-tuning to new tasks. The classification approaches are as follows: (i) feature extraction for both labeled and unlabeled data using the aforementioned pretrained architectures and then k-Nearest Neighbour classification predictions for the unlabeled images (FE/kNN), and (ii) plain category prediction for the unlabeled data using the models trained on the labeled data (Prediction). Data preprocessing includes resizing, normalizing, random horizontal and vertical flipping. Also, we split the data into training and validation sets (80–20%).

To assess the accuracy of the weak labeling process, we visually inspected a random sample of 500 images for each of the four classes and each classification model to estimate the error rates per category and overall. The results of this evaluation are shown in Table 3. Overall, we can see that in the Screenshots and Text out of Image categories, the ViT (Prediction) and the simple CNN (FE/kNN) exhibit relatively low classification errors. Most errors are encountered in the Object Labeling category, with the error rate consistently above 10%. In all categories, the method that yielded the fewest errors in image classification was ViT (Prediction), achieving an error rate of 7.35%. Therefore, the separation of the dataset for creating the set used for addressing the meme image recognition problem in the next section was conducted based on ViT (Prediction).

## 4. Experimental Setup

### 4.1. Class Imbalance and Regular Images Dataset

Using the weak labeling process described in the previous section, we were able to divide the set of the initial ∼27 k images into four classes. To achieve a balance among these classes, random images were removed from the larger classes. However, it should be noted that the smallest category was IM, and to avoid reducing the dataset too much, images from the FBHM dataset, which exclusively contains Image Macros, were added. The number of images added was adjusted to match the size of the second smallest category.

At this point, it was necessary to find an appropriate dataset of regular images and for this purpose, we considered three different datasets: the Microsoft-Common Objects in Context (MS-COCO) [28], the Google’s Conceptual Captions [29], and the ICDAR2019 Robust Reading Challenge—Multi-lingual scene (RRC-MLT-2019) [30]. Using these datasets, we created a mixture of regular images with a size equal to that of the meme dataset. In particular, four mixtures were tried according to the compositions of Table 4. However, before combining the two datasets into one, each subset (meme and regular images) was divided into training, validation, and test sets with an 80%, 10%, and 10% split.

### 4.2. Baseline Models for Meme Detection

We conducted the meme detection experiments using again the CNN, VGG16, ResNet152, EfficientNetB4, and ViT models. Except CNN, all models are initially pre-trained on ImageNet. Their classification head consists of a fully connected network with two ReLU activated hidden layers (1024 and 512 neurons, respectively) with dropout and a third sigmoid activated layer with one neuron. For these models, an exploration with respect to the values for some of their training hyperparameters was first conducted. The tuned parameters include the learning rate, and dropout probability as shown in Table 5. In the case of the ResNet152, the number of frozen layers was also tuned, and in the case of ViT, the same happened for the weight decay parameter.

Data pre-processing involves resizing images to 224×224, and the image values are normalized based on the mean and standard deviation of the ImageNet dataset. The objective function we consider is the binary cross-entropy loss function, and the optimizer is Adam. Additionally, a learning rate scheduling method is applied, reducing the learning rate of the network by one order of magnitude every 5 epochs. Both models are trained for a total of 20 epochs. However, early stopping with a patience of 5 epochs is applied based on the validation set error. The weights of the trained networks, stored at the end of training, are those corresponding to the model that achieved the lowest error on the validation set. Finally, these models were used to analyze their performance on the held-out test set to obtain a comprehensive understanding of the generalization they can achieve.

### 4.3. Evaluation Metrics

For the evaluation of models on the meme detection task, we consider the following performance metrics: accuracy, balanced accuracy, F1 score, area under the ROC curve (AUC).

## 5. Results

The ViT model provided weak labels for 23,171 of these images, as shown in Table 6. The smallest category was the IM category with a total of 2706 images. However, as mentioned, additional images can easily be added to this category from the FBHM dataset, which includes only IM. Looking at the second smallest category, it is the object labeling category with 7205 images. Therefore, 4499 random images from the FBHM dataset were added to the IM category, while random images were removed from the other two categories to reach 7205 images per category.

After finding the optimal hyperparameters for each case, the final training of the models was performed for each of them. Therefore, for each data set mixture, both of the specified architectures were trained. A critical component of this training was the use of the early stopping method. In the initial tests, early stopping was based on monitoring the accuracy of the validation set. This means that training would stop if the accuracy of this set did not improve within 5 epochs. However, when evaluating the trained models on the test sets, their performance was found to be very low. Additionally, during the training, it was observed that the error on the validation set reached a lower bound and started to increase in subsequent epochs without significantly affecting accuracy.

Due to these reasons, it was eventually decided to use early stopping based on monitoring the error of the validation set. By storing the weights that resulted in the lowest error on the validation set, the performance of the models improved, as shown in Table 7. From this validation, it can be seen that model performance is at very high levels, and they can easily distinguish memes from regular images. Finally, it is clear that the use of text within the images, as in the case of [14], does not provide a significant advantage since meme images have distinctive characteristics on their own, as demonstrated by the performance of the MemeTector model [7], which only utilizes the image, similar to the approach taken in this work.

## 6. Conclusions and Limitations

In this work, we identified several categories of meme imagery, and collected a representative diverse meme dataset from Reddit. The collected images needed to be annotated with the corresponding category; thus, two different weak labeling approaches were developed, based on CNN and Transformer architectures. After visually evaluating the predictions and selecting the best method and model, the unlabeled images of the dataset were automatically annotated. The ViT model managed to outperform CNNs and was able to provide high-accuracy distinction. Then, after we alleviated class imbalance, a combination of three existing datasets containing regular images was considered to train models for meme detection resulting in very high accuracy levels.

While PolyMeme is currently one of the most diverse meme datasets in terms of meme format, meme images are constantly changing and new forms emerge that need to be recognized and distinguished by the scientific community. The category of funny images described in this work is one of these forms, and the inability to find such images led to its rejection in this analysis. Therefore, it is necessary to develop models that are constantly updated and informed about developments in the meme space to recognize them as accurately as possible. Finally, other formats exist, such as videos, GIFs, and written text and the rapid growth of social media platforms, especially TikTok, discussed in [25], has highlighted the need for managing video memes, which call for further research and data collection work in this area.

## Figures and Tables

**Figure 1 sensors-24-05456-f001:**
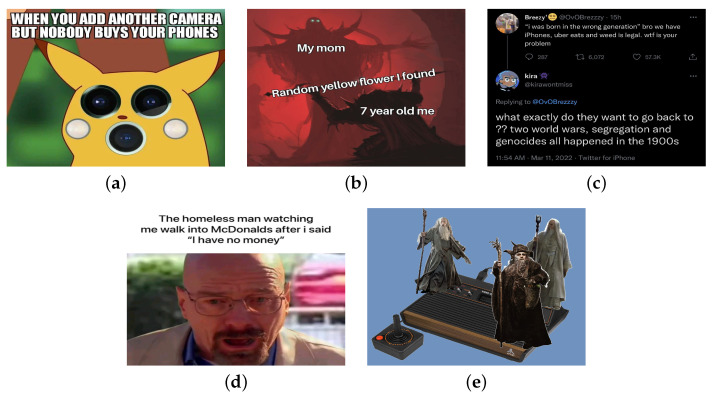
Examples from five popular meme categories in order: (**a**) Image Macro, (**b**) Object Labeling, (**c**) Screenshot, (**d**) Text out of Image and (**e**) Funny Image.

**Table 1 sensors-24-05456-t001:** Existing datasets, their size, the number of different meme categories they contain, whether the samples are annotated with this information, and their source.

Dataset	Size	Categories	Annotation	Source
Facebook Hateful Memes [4]	10,000	1	✗	constructed
Memotion 7 k [9]	10,000	3	✗	Web
MultiOFF [5]	749	4	✗	Social media
DankMemes [6]	1000	4	✗	Instagram
MemeCap [10]	6387	4	✗	Reddit
PolyMeme	26,824	4	✓	Reddit

**Table 2 sensors-24-05456-t002:** The distribution of the sample into the five categories, multiple categories, no categories and cases where the image is not a meme.

Category	No. Images (out of 500)	Percentage (%)
Image Macro	48	9.6
Object labeling	104	20.8
Screenshot	174	34.8
Text out of Image	130	26.0
Funny Image	2	0.4
Multiple categories	25	5.0
No category	7	1.4
Noise	10	2.0

**Table 3 sensors-24-05456-t003:** The error rates that occurred during the visual inspection.

	Model	Image Macros	Object Labeling	Screenshots	Text out of Image	Mean
FE/kNN	CNN	**7.0**	18.2	13.6	**3.2**	10.50
	VGG16	10.0	14.2	5.4	7.4	9.25
	ResNet152	19.0	11.8	4.2	11.2	11.55
	EfficientNetB4	14.2	**11.0**	5.0	9.6	9.95
	ViT	8.8	14.6	4.0	9.6	9.20
Prediction	CNN	8.2	14.8	3.6	9.6	9.05
	VGG16	9.0	11.8	4.0	7.5	8.15
	ResNet152	18.4	12.6	6.8	9.2	11.75
	EfficientNetB4	22.6	13.4	6.6	6.4	12.25
	**ViT**	7.4	12.6	**3.0**	6.4	**7.35**

**Table 4 sensors-24-05456-t004:** The four cases of dataset proportions when mixing the three datasets of regular images.

Case	COCO	CC	ICDAR
A	50.0	25.0	25.0
B	25.0	50.0	25.0
C	25.0	25.0	50.0
D	33.3	33.3	33.3

**Table 5 sensors-24-05456-t005:** Hyperparameters’ pool of values.

Hyperparameter	Values
Learning rate	$10^{-3}$, $10^{-4}$
Dropout	0.60, 0.75
No. layers frozen (ResNet)	10, 35
Weight decay (ViT)	$10^{-3}$, $10^{-4}$

**Table 6 sensors-24-05456-t006:** The categorization of the unlabeled images alongside with the manually annotated ones.

Category	No. Weak Labels	Total Images
Image Macro	2053	2706
Object Labelling	6205	7206
Screenshot	7748	8748
Text out of Image	7165	8164
Total	23,171	26,824

**Table 7 sensors-24-05456-t007:** Model performance (test set accuracy, balanced accuracy, F1 score, and AUC) for each mixing case and architecture. * Accuracy and balanced accuracy scores coincide due to equal class sizes.

Model	A	B	C	D
accuracy/balanced accuracy *				
CNN	96.89	96.58	96.52	96.52
VGG16	98.35	97.68	98.23	98.20
ResNet152	98.77	98.53	**98.92**	98.63
EfficientNetB4	95.09	93.88	94.78	94.63
ViT	98.42	98.37	98.34	98.20
accuracy/balanced accuracy *				
F1 score				
CNN	96.89	96.58	96.52	96.54
VGG16	98.36	97.66	98.22	98.19
ResNet152	98.77	98.53	**98.93**	98.63
EfficientNetB4	95.12	93.91	94.78	94.65
ViT	98.43	98.37	98.34	98.20
AUC				
CNN	99.06	98.87	99.14	99.01
VGG16	99.69	99.58	99.76	99.68
ResNet152	99.81	99.81	**99.88**	99.78
EfficientNetB4	98.68	98.19	98.70	98.54
ViT	99.83	99.81	99.82	99.74

## Data Availability

The data presented in this study are available on request from the corresponding author.

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
