# Peer review of "PolyMeme: Fine-Grained Internet Meme Sensing"

_sensors, 2024, doi:10.3390/s24175456_

Round 1
Reviewer 1 Report
Comments and Suggestions for Authors
This paper presents a new dataset for meme classification. they created a dataset with Reddit images that includes 4 categories and around 27k images. They used a small subset of the data to manually label and trained several models to weakly annotate the rest of the dataset. This, of course, results in suboptimal annotation.
At this point the authors used 3 other well-known datasets to train a model and classify memes. the paper is a bit hazy. not mentioning the parameters used. Also, we noted the dataset is imbalanced and reporting accuracy may not give the best picture regarding the performance of the models. I think the reporting needs to be a bit more in depth.
Author Response
Comments 1: At this point the authors used 3 other well-known datasets to train a model and classify memes. the paper is a bit hazy. not mentioning the parameters used.
Response 1: We have updated subsection 4.2 by adding information with respect to the hyperparameter grid we consider for tuning, model checkpointing, ImageNet pre-training of the backbones, data pre-processing, learning-rate scheduling, classification head architecture, loss function, and optimizer.
Comments 2: Also, we noted the dataset is imbalanced and reporting accuracy may not give the best picture regarding the performance of the models. I think the reporting needs to be a bit more in depth.
Response 2: As stated at line 183 of the submitted manuscript, the size of the regular images class is equal to that of the meme class, thus the binary classification task is balanced and the accuracy metric is capable of reflecting the model’s performance. However, based on the reviewer’s suggestion and for comprehensiveness we have added the following metrics to Table 6 (Table 7 in the revised manuscript): balanced accuracy, F1 score, and AUC. Additionally, we added subsection 4.3 describing the evaluation protocol.
Reviewer 2 Report
Comments and Suggestions for Authors
The creation of a database for the categorisation of meme formats may possibly lead to improvements in the comprehensiveness and accuracy of models in meme classification, which is important in the context of social media content monitoring.
Weaknesses and recommendations:
Size of the dataset: Does not obtain justification for creating such a large sample base (27k images), as there are techniques such as transfer learning and deep pattern matching that could reduce the need for such a large number of images in the training set.
[to be completed:] justify why it was decided to collect such a large dataset;
Compatibility with the purpose of the study: In the introduction, the authors mention that meme classification will catch harmful content such as hate speech and misinformation, but the proposed solution only assessed how the CNN and ViT models perform in classifying images into four types of formats.
[to be completed:] justify how the categorization of meme formats contributes to the detection of harmful content in images and from context;
Lack of details: Lack of precise numbers for the number of samples per category and lack of details about the data cleaning process.
[to be completed:] provide exact numbers of images in each category and details of the cleansing process, including numbers of duplicates removed and images that do not fit the category;
Choice of models: No justification for the choice of specific pre-trained models
[to be completed:] explain why such models were chosen;
Testing of other models: Tests for each mixing case and architecture were carried out for the ViT and ResNet models only.
[additional experiments to be performed:] extend the experiments to include the other models highlighted in the paper;
Concluding remarks: The relevance of meme type may be marginal for modern multimodal models such as ChatGPT-4, which can process both text and images. The paper makes a valuable contribution to the field of meme analysis, especially in the context of format diversity and the use of advanced semi-automatic annotation methods. However, clarifying the choice of methods, providing detailed data and extending the tests to other models could significantly enhance the quality and reliability of the study.
Author Response
Comments 1: Size of the dataset: Does not obtain justification for creating such a large sample base (27k images), as there are techniques such as transfer learning and deep pattern matching that could reduce the need for such a large number of images in the training set. [to be completed:] justify why it was decided to collect such a large dataset;
Response 1: The following justification is provided in the Introduction section: “Additionally, memes are special image cases involving subtle differences in terms of the background image semantics as well as the overlay text font appearance and position. These peculiar cues that differentiate them from the images of standard image recognition tasks, render simple transfer learning less powerful and backbone fine-tuning is deemed beneficial for the task of meme detection and classification. Hence, the datasets used for such tasks should be of large size in order to enable the backbones to learn the corresponding patterns and adequately generalize, however current available datasets are of limited size as shown in Table 1.”
Comments 2: Compatibility with the purpose of the study: In the introduction, the authors mention that meme classification will catch harmful content such as hate speech and misinformation, but the proposed solution only assessed how the CNN and ViT models perform in classifying images into four types of formats. [to be completed:] justify how the categorization of meme formats contributes to the detection of harmful content in images and from context;
Response 2: Automatic online meme sensing (discriminating memes from regular images in the web) enables a second step of meme analysis/understanding with the aim to classify it as harmful or not and then if a flag is raised a human moderator can be employed to provide a final decision. In this paper, we address the first step of meme detection. Now, considering many types of image memes (instead of using only image macros) makes the trained models robust to all possible cases. We have added a clarifying parentheses in the first paragraph of the Introduction: “To this end, a first necessary step involves robust (to all meme types) methods for ``sensing'' meme content”.
Comments 3: Lack of details: Lack of precise numbers for the number of samples per category and lack of details about the data cleaning process. [to be completed:] provide exact numbers of images in each category and details of the cleansing process, including numbers of duplicates removed and images that do not fit the category;
Response 3: Table 5 (Table 6 in the revised manuscript), provides the meme category sizes. In section 3.3, we clearly state that we removed 189 exact duplicates, 223 near duplicates, and 249 images without overlay text. Regarding the discarded class of funny images we report in the revised manuscript (Section 3.3) that its size was 349 images.
Comments 4: Choice of models: No justification for the choice of specific pre-trained models [to be completed:] explain why such models were chosen;
Response 4: We have added the following in Section 3.4: “We have made these architecture choices based on the fact that they have demonstrated state-of-the art performance on numerous computer vision tasks and have proven to be easily adaptable through fine-tuning to new tasks.”
Comments 5: Testing of other models: Tests for each mixing case and architecture were carried out for the ViT and ResNet models only. [additional experiments to be performed:] extend the experiments to include the other models highlighted in the paper;
Response 5: We have conducted the corresponding experiments and in the revised manuscript (Table 6 in the initially submitted manuscript - Table 7 in the revised manuscript) we provide results also for the CNN, VGG16, and EfficientNetB4 architectures.
Round 2
Reviewer 2 Report
Comments and Suggestions for Authors
All corrections have been made, and the article has been accepted in recognition of the effort put into it. However, the overall topic and the general result are not the best in terms of advances in the field of image classification using convolutional neural networks. Moreover, in the field of machine learning, the approach is not particularly innovative or progressive.